# How to Improve Patient Safety Literacy?

**DOI:** 10.3390/ijerph17197308

**Published:** 2020-10-07

**Authors:** Yoon-Sook Kim, Hyun Ah Kim, Moon-Sook Kim, Hyuo Sun Kim, Mi Jeong Kwak, Jahae Chun, Jee-In Hwang, Hyeran Kim

**Affiliations:** 1Department of Quality Improvement, Konkuk University Medical Center, Seoul 05030, Korea; 20050107@kuh.ac.kr; 2Office of Quality Innovation, Samsung Medical Center, Seoul 06351, Korea; hyunakid@gmail.com; 3Medical Nursing Department, Seoul National University Hospital, Seoul 03080, Korea; behappynow@snuh.org; 4Performance Improvement Team, Uijeongbu St. Mary’s Hospital, Uijeongbu 11765, Korea; bjpkhsnr@gmail.com; 5Quality Improvement Team, Korea University Anam Hospital, Seoul 02841, Korea; Kuku105@hanmail.net; 6Nursing Department, Severance Hospital, Seoul 03722, Korea; jhchun@yuhs.ac; 7Department of Nursing, Kyung Hee University, Seoul 02447, Korea; jihwang@khu.ac.kr

**Keywords:** patient safety, literacy, patient and family education

## Abstract

The aim of this comparative study involving pre- and post-tests was to analyze the effectiveness of patient safety educational materials developed for the Comprehensive Plans for Patient Safety in Korea (2018–2022), and to suggest how to improve patient safety literacy. A face-to-face survey interview comprising items related to general information and patient safety literacy was completed by 217 patients and their families who visited three general hospitals in Seoul and one general hospital in Gyeonggi-do for treatment between 25 October and 15 November 2019. In the interview, the patients were asked questions about whether the patient safety educational materials were “easy to understand,” provided “help in safe hospitalization,” and enabled patients to practice patient safety independently (“do it yourself”). The literacy of the patient safety educational materials was analyzed using a paired *t*-test with a *p* value of 0.05. The comparison between patient safety literacy on pre- and post-tests revealed that among all participants, there were significant differences in “easy to understand,” “help in safe hospitalization,” and “do it yourself” scores. To improve patient safety literacy, patient education materials need to optimize communication by improving patients’ knowledge, skills, and attitudes for maintaining and promoting healthy living.

## 1. Introduction

### 1.1. What Is Health Literacy?

Health literacy is defined as “the degree to which individuals have the capacity to obtain, process, and understand basic health information and services needed to make appropriate health decisions” according to Healthy People 2020. In Healthy People 2030, health literacy definition is not only applicable to individuals but also to organizations that create and provide health-related information and services. Personal health literacy is defined as the degree to which an individual is able to find, understand, and use information and services to make informed health-related decisions and actions for themselves and others. This definition emphasizes the individual’s ability to use health information rather than only understand it. Organizational health literacy is the degree to which an organization equitably supports individuals to find, understand, and use information and services to make informed health-related decisions and actions for themselves and others. In other words, the organization is responsible for providing health literacy by researching and developing health-related information that is easy to understand, dependable, and usable by healthcare consumers [1]. 

It is known that health literacy not only affects individuals’ perception and management of health status but also their health care activities such as regular checkups, vaccinations, and medicine intake [2]. However, many people do not clearly know how to manage their health and have difficulties in understanding health information such as doctors’ prescriptions, precautions related to surgery and examination, medication instructions, diet advice, and health education materials [3]. 

### 1.2. Impact of Health Literacy on Patient Safety

Studies have shown that people with limited health literacy are less likely to undergo health screenings and comply with medical orders; hence, they are more likely to experience delayed diagnosis of medical problems, have difficulty managing their medical condition(s) and/or those of their children, and have a greater risk of mortality [4,5,6,7]. In addition, studies have shown that people with limited health literacy have more experiences that threaten patient safety, such as medication error [8,9]. Therefore, patient safety highly depends on whether health care information is properly delivered to the patients and is properly understood by them.

Health literacy is at the core of patient safety planning and activities [8]. Specifically, miscommunication and lack of information (such as not providing patients with information about the surgical site or prescription drugs) are the most common causes of medical errors, including miscommunication among the medical staff, resulting in wrong medication administration or surgery at the wrong site. Such misunderstandings often occur between patients and healthcare providers [10].

Recently, self-care needs of patients have been increasing due to a decrease in the length of their hospital stay; this is accompanied by an increase in polypharmacy and the prevalence of chronic diseases. Therefore, the importance of accurate communication between patients and healthcare providers has also increased [11,12,13]. 

### 1.3. How to Improve Health Literacy for Patient Safety?

Patients who recognize that accurate information is an important aspect of patient safety can actively participate in the care decision-making process [14,15]. Patients need to have access to reliable information to participate in their care programs and medical staff also requires educational materials to advise patients on the same [14]. It is also necessary to improve the readability of these educational materials. Educational brochures, instructions on prescription drugs, and flyers can help patients and their families maintain their health knowledge after leaving the hospital. In some cases, written health information may be the only way patients and their families learn about important health guidelines [16].

The Comprehensive Plan for Patient Safety in Korea (2018–2022) was established as part of the National Patient Safety Culture Project. The plan includes (1) development of educational materials to help patients participate in their care and decision-making, (2) patient engagement campaign, and (3) patient safety culture campaign related to “the right to know the patient” [17]. As part of this plan, we developed patient safety educational material for patients and their families and also measured patient safety literacy. Only one of the studies in the recent literature has focused on the relationship between health literacy and patient safety; however, no study has taken into account patient safety literacy. 

Since literacy was considered important for patient safety, in this study, patient safety literacy was defined as “the level of readability, understanding, and actionability needed for information related to patient safety in order to ensure the same and minimize risks.”

The purpose of this study was to investigate the effectiveness of the existing patient safety educational materials in Korea and provide suggestions to improve patient safety literacy.

## 2. Materials and Methods 

### 2.1. Research Design

This is a comparative study involving pre- and post-tests, which compared the effectiveness of educational materials for patient safety. The interventions used in this study included developed educational materials for patient safety. 

### 2.2. Participants and Data Collection

The survey was conducted on patients and their families who visited three general hospitals in Seoul and one general hospital in Gyeonggi-do for medical treatment. A total of 230 patients or families answered the survey and data from 217 participants were used for the analysis, excluding 13 patients who did not complete the survey. Study participants were inpatients, outpatients, and their families. The survey was conducted from 25 October to 15 November 2019. 

*Ethical statement*: All subjects gave their informed written consent for inclusion before they participated in the study. The study was conducted in accordance with the Declaration of Helsinki, and the protocol was approved by the Ethics Committee of Konkuk University Hospital (KUH 2019-10-022). 

After receiving written consent from the participants, the research assistants who were trained in using the structured questionnaire conducted a face-to-face survey. After explaining what the consent entailed, the research assistants confirmed that the participants understood the same by repeatedly explaining it until no question was asked. The questionnaire comprised items on general information and patient safety literacy. The average time to complete the questionnaire was about 20 minutes. The survey was conducted as shown in Figure 1.

### 2.3. Development of Patient Safety Educational Materials

In accordance with the Comprehensive Plans for Patient Safety [17], we developed educational materials that could help patients and their families engage in the prevention of medical errors. For the development of such materials, we conducted a web search (patient safety-related overseas institutions) [18], FGIs (focused group interviews) [14], hospitalization guidelines analysis, and took expert advice. 

In Step 1, we reviewed patient safety educational materials (brochures, videos, and flyers) obtained from four patient safety-related overseas institutions. The scope of the web search was outlined through a meeting of researchers and advisors at WHO (World Health Organization); patient safety-related institutions; and top five hospitals in the United States, Canada, the United Kingdom, Australia, Taiwan, and Singapore.

The survey included contents and methods related to patient safety education provided by each institution and hospital. Our researchers mainly referred to materials from four institutions for developing the patient safety education materials. 

The Agency for Healthcare Research and Quality provides free resources that can be used by medical staff to effectively communicate questions and concerns with patients and caregivers in limited time, when seeing patients in a hospital, and primary care or other healthcare settings [18]. 

The Institute develops and shares tools for improving quality of patient safety and creating a culture of patient safety within healthcare institutions. In addition, it provides useful resources for experts and networks, as well as patients and caregivers [18]. 

Canadian Patient Safety Institute’s patient and family educational content has been made available to the general public through guides, tools/data postings, and related links [18]. 

In 2006, WHO recognized the importance of the role of patients and caregivers in patient safety and formed “Patients for patient safety” to promote patient and caregiver participation in patient safety activities. “Patients for patient safety” prioritizes patient self-care decisions, voice of patients and families, and seeks to establish partnerships between health professionals, patients, families, and communities [18].

In Step 2, FGIs were conducted, which involved 11 patients/families and 9 patient safety officers who had received or experienced patient safety education. The FGIs were conducted according to the interview questionnaire prepared in advance. Four researchers were present, two as interviewers, one as a warrior, and one as an observer. The interview lasted about 2 hours, and the contents of the interview were tape recorded; it consisted of opening, introduction, transition, main, and closing questions. FGIs were analyzed using a content analysis method. The key statements of the FGIs are as following.

#### 2.3.1. Patients/Families

“I have received a lot of education from doctors and nurses in hospitals. However, after the training I can’t remember more than 80 % of the education provided by the medical staff.”“I don’t know what the doctor or nurse is saying or what patient safety is.”“When I go to the hospital, the doctor or nurse tells me what they want to say and don’t listen to me.”“I write down the questions in advance and ask the doctor. My body must be taken care of by me.”“I know my health condition and it’s important for me to tell the doctor about it. However, the doctor only talks and does not listen to me.”

#### 2.3.2. Patient Safety Officers

“The age of unilateral education for patients seems to be over. The materials of patient safety education in Korea are too negative. Only irrelevant information is available in it. Such education does not guarantee patient safety.”“It seems important to advise patients to ask questions from doctors and nurses.”“Since there is so much education that patients need to receive, it is important to selectively conduct the necessary education at the same level every day.”“There should be a provision that allows patients to ask questions freely when they feel anxious.”“At the national level, campaigns should be created to develop a culture that can change the perceptions of patients, families, and staff in hospitals.”
In Step 3, we analyzed the hospitalization guidelines of 14 hospitals (six tertiary hospitals, five general hospitals, and three long-term care hospitals). The summary of the hospitalization guidelines is as follows, and the contents related to patient safety are marked with “S.”
Things to prepare for admission;Time for doctors’ rounds and nurse patrol time;Patient identification method (S), fall prevention method (S), pain management, and hand hygiene (S);Guidance on preparation for hospitalization of pediatric patients;Informing about medicines before admission (S);Inpatient medical guardian appointment service, discharge procedure, application for copies of medical records, patients’ rights and obligations, voice of customer, social work, privacy protection request, and medical expense inquiry;Hospital room etiquette, mealtime, visit time, hospital facilities (including facilities for people with disabilities), fire prevention and evacuation tips (S), no smoking (S), prevention of theft, religious office, and parking fee.
In Step 4, we developed the patient safety educational materials based on the data collected in Steps 1–3 and the advice of 14 external experts: patient safety and quality improvement team leaders, professor of nursing college, head nurse in general unit, and nurse manager of internal medicine. 

Our researchers considered the following points while developing the patient safety educational materials: (1) emphasis on “The patient is at the center of patient safety” (“Do not talk about me without me”), (2) assurance of patient safety through patients, and (3) usage of simple and clear terms (bring, speak, question, and check).

We revised the patient safety educational materials according to expert opinions (Figure 2).

### 2.4. Development of Patient Safety Literacy Tool

We conducted systematic reviews and considered expert advice to develop patient safety literacy tools. First, we conducted a systematic review; the results are as follows:Key question: What tools are available to assess health literacy (or patient safety literacy)?Search engines: PubMed, Embase, Cochrane Database, KoreaMed, Korean Studies Information Service System, Korean Medical Database.Research methods: Full-text accessible paper (original study, evidence-based guideline, systematic review).Period: Unlimited (from the first publication on health literacy until 12 July 12 2019).Languages: English or Korean.Study subjects: Over 18 years old.Exclusion criteria: Studies focusing on health care providers (doctors, nurses, pharmacists, therapists, etc.) and teachers (teachers, professors, etc.).A total of 4447 documents were searched, and a total of seven health literacy tools were extracted, excluding duplicate or wrong topics.

Second, we reviewed the health literacy tools identified through the systematic review and selected those that could be applied to patient safety literacy (Figure 3). 

Among the seven health literacy tools, Newnest Vital Sign was found to evaluate the understanding of nutritional information; hence, it was excluded because it is unrelated to safety literacy. Seven internal researchers, comprising patient safety managers, nursing directors, and nursing professors, scored the applicability of the remaining six health literacy tools in patient safety literacy. The scores ranged from one (very difficult to be applicable) to five points (very likely to be applicable).

The results evaluated by the seven experts are presented in Table 1. The Screening Questions for Limited Health Literacy tool, which had the highest total score (32 points), were selected. This tool has five response options: always, often, sometimes, occasionally, and never. Choosing any one of the three (“sometimes,” “occasionally,” or “never”) indicates limited health literacy. 

Third, upon the advice of 14 experts, “Screening Questions for Limited Health Literacy” was modified and used as a tool to measure patient safety literacy. The experts included patient safety and quality improvement team leaders, professor of nursing college, head nurse in general unit, and nurse manager of internal medicine. 

The patient safety literacy tool consisted of three questions: Were the patient safety educational materials (words, sentences, meanings, etc.) easy for you (the patients and their families) to understand? (“Easy to understand”).Did the patient safety educational materials help ensure safe hospitalization? (“Help in safe hospitalization”).Can you practice patient safety yourself without the help of others? (“Do it yourself”).

The patient safety literacy tool had four response options: “always,” “often,” “occasionally,” and “never.” If any one of the two options of “occasionally” or “never” was chosen, it indicated limited patient safety literacy. 

### 2.5. Data Analysis 

The literacy of the patient safety educational materials was analyzed by a paired *t*-test, and the statistical significance of the differences between the groups was determined based on a *p* value of 0.05. Data analysis was performed using PASW software (version 24.0; SPSS Inc., Chicago, IL). We also analyzed differences between the responses of patients and families. Before this analysis, we conducted a homogeneity test using chi-square because differences in general characteristics of the two groups could have affected the results of the study. The general characteristics of the study participants and the degree of patient safety literacy were presented as real numbers, mean, and standard deviation.

## 3. Results

### 3.1. Homogeneity Test

The results of the homogeneity test between patients and families are shown in Table 2. Age, education level, and patient safety education received from medical staff were not statistically significant between patients and families but were statistically significant by gender (*p* = 0.005). By gender, females comprised 51.2% of the patients and 70.2% of the families. Regarding age, most patients were over 61 years at 32.5%, while for families most were aged 31–40 years at 26.6%. The education level of the patients indicated that 7.3% were elementary school educated or below, 9.8% completed middle school, 36.6% completed high school and college, and 9.8% held master’s degree or higher. As for the education level of families, 6.4% graduated from elementary school or below, 8.5% were middle school graduates, 22.3% were high school graduates, 51.1% completed college, and 11.7% had master’s or higher degree. The patient safety education contents received by the patients from the medical staff included the following: (a) Tell your medicines (69.9%), (b) Orientation to hospitalization (68.3%), (c) Tell your health condition (67.5%), (d) Prevent falls (57.5%), (e) Handwashing methods (55.3%), (f) How to participate in health-related decision-making? (48.0%), and (g) Prepare a list to ask your doctor (19.5%). The patient safety education contents received by the families from the medical staff included the following: (a) Orientation to hospitalization (67.0%), (b) Tell your medicines (64.9%), (c) Tell your health condition (61.7%), (d) Handwashing methods (48.9%), (e) Prevent falls (42.5%), (f) How to participate in health-related decision-making? (38.3%), and (g) Prepare a list to ask your doctor (16.0%).

### 3.2. Comparisons between Pre- and Post-Patient Safety Literacy Tests

Table 3 shows the comparison of patient safety literacy about patient safety educational materials based on pre- and post-tests. 

Among all participants, there was a significant difference in “Easy to understand” between pre- (3.05, SD = 0.55) and post-tests (3.17, SD = 0.64) (*p* = 0.006). There was a significant difference in “Help in safe hospitalization” scores between pre- (2.85, SD = 0.63) and post-tests (3.20, SD = 0.47) (*p* < 0.001), as well as in the “Do it yourself” scores between pre- (3.35, SD = 0.80) and post-tests (3.49, SD = 0.76) (*p* = 0.004).

Among the patients, there were significant differences in the “Easy to understand” scores between pre- (3.02, SD = 0.57) and post-tests (3.15, SD = 0.68) (*p* = 0.017). There were also significant differences in the “Help in safe hospitalization” scores between pre- (2.84, SD = 0.63) and post-tests (3.17, SD = 0.49) (*p* < 0.001). Finally, significant differences were found in the “Do it yourself” scores between pre- (3.37, SD = 0.76) and post-tests (3.50, SD = 0.80) (*p* = 0.038). 

Among the families, there was a statistically significant difference in the “Help in safe hospitalization” scores between pre- (2.86, SD = 0.63) and post-tests (3.24, SD = 0.85) (*p* < 0.001). 

## 4. Discussion

In this study, patient safety literacy tool and patient safety education materials were developed, and the effect of patient safety education materials was identified as patient safety literacy. We attempted to present the importance of patient safety education materials to improve patient safety literacy. This study is significant because for the first time, patient safety literacy skills were introduced worldwide and were used to evaluate the effectiveness of patient safety education materials.

The results of this study revealed that the education level of elementary school or below was 7.3% for patients and 6.4% for families. A previous study that surveyed the health information literacy of adult inpatients showed education level as above 61.8% [19]. In most studies, education level was found to be a related factor of health information literacy and low education level showed low health information literacy [19,20,21], but high education level did not always show positive correlation with high health information literacy [22]. Unlike previous health literacy studies, even if the subjects of this study had a high level of education, it was difficult to identify the positive or negative effects on patient safety literacy. Perhaps, more research is needed in this direction.

The most frequently received patient safety education by the study participants was “Tell your medicines” (69.9% of the patients) and “Orientation to hospitalization” (67.0% of the families). In other studies, the most frequent patient safety education was “Describe your medical condition” in 69.0% of the patients and “Hospitalization orientation” in 62.0% of the families [23]. When a patient is hospitalized in Korea, one of the family members lives in the hospital with the patient and takes care of the patient. For this reason, it was thought that medical staff should provide more “Orientation to hospitalization” education to the families than to the patients.

In Korea, Patient Safety Offer conducted campaigns such as fall-prevention education and “Speak Up” for patient safety education. However, it has been reported that such patient safety education has limitations in preventing medical errors [14]. Therefore, it is necessary to develop standardized patient safety education materials that can help prevent medical errors in Korea. For this purpose, we developed patient safety education materials (“4 Tips for patient safety”) and patient safety literacy. In the study results, the newly developed patient safety education materials showed higher scores in “Easy to understand,” “Help in safe hospitalization,” and “Do it yourself” than existing patient safety education materials. It is believed that the newly developed patient safety education material will be helpful in promoting patient safety.

Although there are no prior studies related to patient safety literacy, health literacy has been shown to affect individuals’ daily health care behaviors. Patients with low level of health literacy are not aware of the importance of early detection and treatment of diseases; therefore, it is confirmed that they are less likely to use disease-prevention related services such as vaccination against infectious diseases like influenza or pneumococcal, and regular cancer screening [24]. This finding supports the understanding that patients with high level of patient safety literacy performed high patient safety activities.

In addition, it was reported that patients with low health literacy level were less inclined to ask questions or search for new information, and their communication efficacy decreased [25,26]. This supports the fact that patient safety prevention is helpful as it allows patients to inquire about and inform the medical staff about their conditions and eventually achieve a high level of patient safety literacy.

Patient education is the process of inducing behavioral changes among patients by improving their knowledge, skills, and attitudes necessary to maintain and promote a healthy life. Patient education enhances the potential of patients to make informed decisions regarding their healthcare options, develop self-care management, make healthy lifestyle choices, and participate in ongoing care [27]. The Joint Commission has said that it is necessary for healthcare providers to develop a comprehensive response to health literacy for patient safety and to help patients and families understand and act on their health in better ways. It has also said that in addition to disease-related problems, improving health information materials and their understanding are vital for patient safety [28,29]. This suggests that patient safety education is important and so is developing appropriate and comprehensible patient safety education materials, to ensure easy understanding.

Medical terminologies can be difficult to understand for patients and their families, making it difficult to understand health information. It is important to provide patient-tailored information to ensure patient safety, but it is not easy to do so in Korea, where several patients provide treatment in a short time. Therefore, it is necessary to provide impressive patient safety education materials to them and evaluate their patient safety literacy knowledge and understanding.

## 5. Conclusions

Although patient safety is an important topic worldwide, there is no literature on the subject. No limitations were identified in the development of the patient safety literacy because there has been no previous research on this topic. On the contrary, various research methods (literature and systematic reviews, FGIs, and expert advice) were used to strengthen the validity and reliability of the findings.

The root cause of most events that threaten patient safety is lack of education among patients and families. Although national level and medical institutions provide education to patients and their families, incidents threatening patient safety are not reduced and continue to recur. Medical institutions’ services are utilized by patients of different cultures, experiences, and education levels. Developing and providing tailored educational materials to these patients requires extensive manpower and is expensive as well. Therefore, it is important to develop patient safety educational materials that can be easily taught and understood by anyone.

The patient safety educational materials developed in this study can be easily taught by anyone and can be understood even by patients and families with limited health literacy; these patients and their families can practice patient safety by simply referring to the educational materials provided. In addition, we found that the patient safety educational materials developed in this study improved the safety literacy of patients and their families.

We hope that the results of this study are helpful for healthcare providers striving to develop safety measures for patients. We also expect our findings to be actively used at the national level.

## Figures and Tables

**Figure 1 ijerph-17-07308-f001:**
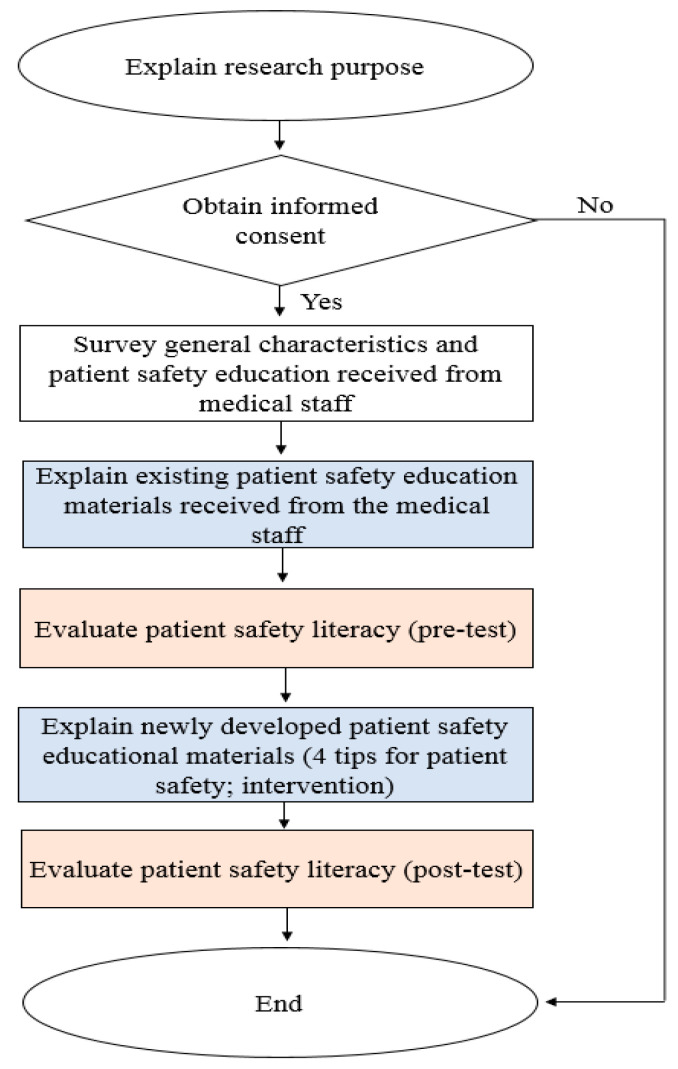
Flow of survey process.

**Figure 2 ijerph-17-07308-f002:**
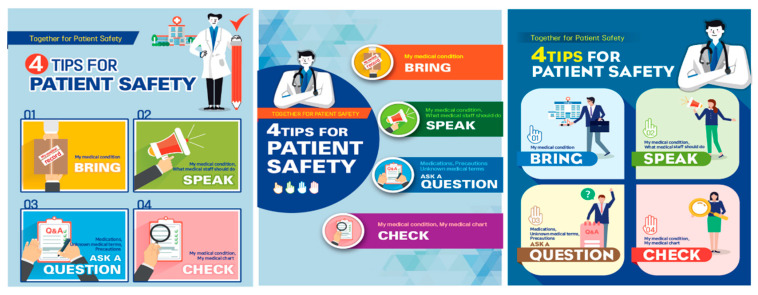
Patient safety educational materials.

**Figure 3 ijerph-17-07308-f003:**
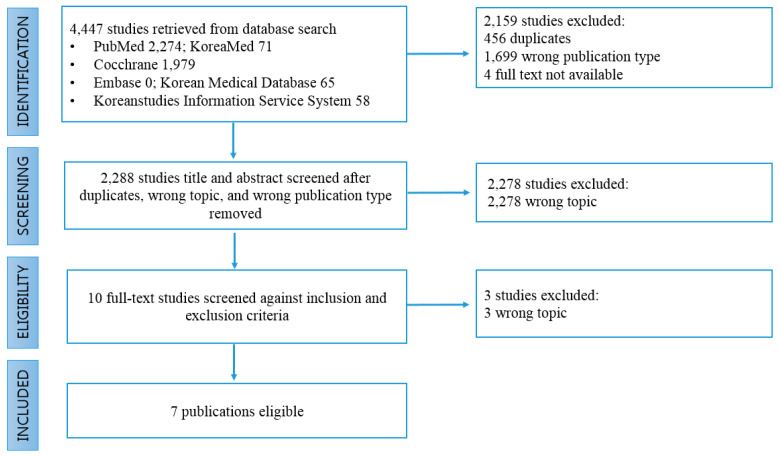
PRISMA flow diagram.

**Table 1 ijerph-17-07308-t001:** Results of evaluation of studies by seven researchers.

Tool	Number of Items	Time ^1^	R1	R2	R3	R4	R5	R6	R7	Applicability (Total)
Rapid Estimate of Adult Literacy in Medicine	66	3–6	4	1	3	3	4	1	1	17
Rapid Estimate of Adult Literacy in Medicine - Revised	8	2	5	4	4	3	4	3	4	27
Test of Functional Health Literacy in Adults	Reading comprehension 50 Numerical ability 17	22–25	3	1	1	4	2	1	1	13
Short Test of Functional Health Literacy in Adults	Reading comprehension 36	12	3	1	2	2	3	1	4	16
Screening Questions for Limited Health Literacy	3	1	3	5	5	5	4	5	5	32
Single Item Literacy Screen	1	2	3	5	4	2	4	1	3	22

^1^ Time indicates the average number of minutes taken to complete the survey.

**Table 2 ijerph-17-07308-t002:** Results of the homogeneity test between patients and families.

Variable	Patient, *n* = 123		Family, *n* = 94		*p* Value
	*N*	%	*N*	%	
Gender					
Female	63	51.2	66	70.2	0.005 **
Male	60	48.8	28	29.8	
Age range (years)					
≤ 30	15	12.2	12	12.8	0.092
31–40	22	17.9	25	26.6	
41–50	23	18.7	24	25.5	
51–60	23	18.7	17	18.1	
≥ 61	40	32.5	16	17.0	
Education level					
Elementary school or below	9	7.3	6	6.4	0.164
Middle school	12	9.8	8	8.5	
High school	45	36.6	21	22.3	
College	45	36.6	48	51.1	
Master’s degree or higher	12	9.8	11	11.7	
Patient safety education received from medical staff					
Handwashing methods (y)	68	55.3	46	48.9	0.353
Preventing falls (y)	96	57.5	71	42.5	0.663
Orientation to hospitalization (y)	84	68.3	63	67.0	0.843
Tell your medicines (y)	86	69.9	61	64.9	0.433
Tell your health condition (y)	83	67.5	58	61.7	0.377
How to participate in health-related decision-making (y)	59	48.0	36	38.3	0.155
Prepare a list to ask your doctor (y)	24	19.5	15	16.0	0.499

** *p* < 0.01; y, yes.

**Table 3 ijerph-17-07308-t003:** Comparisons between patient safety literacy in pre- and post-tests.

Variable	Total, *n* = 217	*p* Value	Patient, *n* = 123	*p* Value	Family, *n* = 94	*p* Value
	Mean (SD)		Mean (SD)		Mean (SD)	
Easy to understand						
Pre	3.05 (0.55)	0.006 **	3.02 (0.57)	0.017 *	3.09 (0.52)	0.151
Post	3.17 (0.64)		3.15 (0.68)		3.18 (0.59)	
Help in safe hospitalization						
Pre	2.85 (0.63)	0.000 ***	2.84 (0.63)	0.000 ***	2.86 (0.63)	0.000 ***
Post	3.20 (0.47)		3.17 (0.49)		3.24 (0.85)	
Do it yourself						
Pre	3.35 (0.80)	0.004 **	3.37 (0.76)	0.038 *	3.32 (0.85)	0.050
Post	3.49 (0.76)		3.50 (0.80)		3.48 (0.71)	

* *p* < 0.05, ** *p* < 0.01, *** *p* < 0.001; y, yes.

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
