# Peer review of "How to Improve Patient Safety Literacy?"

_ijerph, 2020, doi:10.3390/ijerph17197308_

Round 1

Reviewer 1 Report

Thank you. This is a potentially interesting and fertile field of study. There are a number of difficulties with the focus, methods and results presented in this work. The uses of bulet points in lieu fo developed narative is unhelpful.

Line 38 - you may want to explain what you mean by 'self-determination skills' as it may not be self evident to the readership.

Line 40-41. This is really interesting, but what has the greatest impact. Some sources are more reliable than others for example health professionals are likely to provide credible advice and education wheras social media and dare I say, search engines, offer information of a variable quality. In the UK patients access a lot of their information from the internet and social media. The user may be literate in the sense that they can access information, but does that mean they can differentiate between the credible and the unreliable or plainly wrong advice or information.

Line 46 on. You are assuming that literacy, in the form of information is sufficient, but the quality of the information is also critical. See point above.

Line 51 '...have more experiences that threaten patient safety.' Such as?

Line 55. '...miscommunication and lack of information (such as not providing patients with information about 55 the surgical site or prescription drugs) are the most common causes of medical errors...' What evidence do you have to support this - does it relate to citation [8] if so this is not entirely clear.

Line 66. You have introduced 'reliable information' into the narrative. This should be reflecrted in the earlier parts of the narrative.

Line 68. You mention readability then include video. I wonder if accessibility and literacy is more than just the written form. You later define lireracy in terms of readability this is more about information in the written form and not information provided in any other way. This is a key point which you should be very cleart about

Lin 95. You mention subjects providing written concent - given the nature of the study how sure are you that they understood what they were consenting to? This needs an explanation in the narrative.

Table 1. This needs a detailed search strategy. What was the aim of the search, what search terms were used, what inclusion and exclusion criteria were used for selecting the web sites/information to include. The table is unclear, and confusing. It is not useful as a summary of data and is text heavy making it difficult to interpret.

Lines 112 - 157 These sections lack clarity and detail. The extensive use of bulleted list is not helpful. This should be developed into a narative section

Line 160. You state that you undertook a systematic review of the literature. This is not reflected in the methods. There is no question, search terms and Boolean operator characteristics. I would also expect to see the filtering of the papewrs using a PRISMA flow diagramme or similar.

Line 182. What qualified the experts as experts. How did they undertake the screening. Was there a test for interrater reliability? Was a likert scale or similar used?

Line 215. Section 3.2 is difficult to follow and interpret given the lack of detail of what is in the pre and post test alongside the measure of literacy and understanding.

The discussion and conclusion are impossible to evaluate given the limitations in the previous sections - sorry.

Table 2. The way in which the data were summarised is not clear. Are the columns R1-R7 populated with the mean of the respondants and then the applicability is the mean of the means. I am not clear how to interpret the data in the table.

Line 193 Easy to understand by who? Are the patients self reporting? If not how can others judge whether the patient finds it easy to understand?

Line 201. Section 2.5 needs much more detail

Line 216. I am not at all clear how you are measuring patient literacy

Author Response

Thank you for the comments and helpful suggestions regarding our manuscript. We have modified the manuscript accordingly and provided the revised attach file.

Reviewer 2 Report

Thanks for the authors contribution to deliver this paper, it has demonstrated the great determination throughout the process, but some of the content suggested by the reviewers are as follows:

In this paper line 80 ‘patient safety literacy was defined as “the level of readability, understanding, and actionability’ involving pre- and post-tests, which compared the effectiveness of developed educational materials for patient safety. Please explain more details about samples included criteria? Since different disease and treatment will affect the patients and families emotional destress and answer, for example and visit for outpatient department and hospitalized all included?! And the test timing? Before and after details?

Line 102, Development of patient safety educational materials section, the complicated four steps and details which can see the authors ambitious attempted to make three papers into one paper, causing each part of them not able to sound scientifically, suggest the authors to revise the systematic review literatures, FGI section of the main paper to make the four steps simpler and clear.

line 112: FGIs were conducted, which involved 11 patients/ families and nine patient safety 113 officers.’, please describe the specific process of FGI conduct time, and key statement how to make? the qualitative data analysis method?

Line 173-4, ‘seven health literacy tools were extracted’, has each material  been promoted before this investigation? whether hospitals staff have been informed?  Did those staff practice those materials? Those also would affect hospital staffs’ performance and additionally to affect the patients outcomes.

How the researcher differential seven materials which one was the most useful? and how 

Line 134-148, the research team checking 14 hospitals’ hospitalization guide there are only 6 materials related to patient’ safety, how the research team to ask the patients and families seven health literacy tools?

Please add the ethical consideration details

Line 251, “4 Tips for patient safety’ did these tips also have been tested in this survey, how did it work? how to measure it works?

Results section content need to consider revised some content for matching the paper's completeness and accuracy purpose.

Discussion section: suggest the author need to rethink this section to make the findings should be appropriate to discussed it in the content with previous literature.

Author Response

(The authors gave the same response as above.)

Reviewer 3 Report

A comment and a detail:

  • Comment: the Discussion now reiterates the motivation for conducting the study. This is fine but a Discussion should also put the research in perspective and mention possible imperfections. For example: the encouraging results of the experiment may partly be influenced by the fact that participants knew beforehand what the purpose of the experiment was (cf. Figure 1). 
  • Detail line 127: "advice" should here be spelled as "advise" (verb, not noun)

Author Response

(The authors gave the same response as above.)

Round 2

Reviewer 1 Report

Thank you.

The paper is improved in terms of readability and utility. Thank you for ther additional work you have undertaken.

Line 40 & 230 should both dates be 2020?

Line 189 refers to experts, are these researches too, as in line 219? Both terms are used in the paper.

Author Response

Thank you for the comments and helpful suggestions regarding our manuscript. We have modified the manuscript accordingly and provided the revised details below.

Reviewer reports:

Reviewer 1: Thank you. The paper is improved in terms of readability and utility. Thank you for ther additional work you have undertaken.

  1. Line 40 & 230 should both dates be 2020?

Response: In Line 40, 'Healthy People 2020' and 'Healthy People 2030' refers to a strategy presented by the U.S. Department of Health and Human Service aimed at encouraging health and well-being for the next 10 years. Health literacy definition changed from 'Healthy People 2030'.

  1. Line 189 refers to experts, are these rese arches too, as in line 219? Both terms are used in the paper.

Response: Line 189 is referring to an external expert, and Line 219 mentions an internal researcher. I have revised the sentence to clarify the meaning.

  • Line 187-189: In Step 4, we developed the patient safety educational materials based on the data collected in Steps 1-3 and the advice of 14 external experts: patient safety and quality improvement team leaders, professor of nursing college, head nurse in general unit, and nurse manager of internal medicine.
  • Line 218-220: Seven internal researchers, comprising patient safety managers, nursing directors, and nursing professors, scored the applicability of the remaining six health literacy tools in patient safety literacy.

Reviewer 2 Report

"Less is more" is the motto of all things, especially for a complex conceptual essay. How to express its meaning in a simple and clear way is crucial for an excellent paper.
Thank the authors for theirs hard work to improve the quality of the content of this article.
This article has its own value for the development of Korean health industry. For the publication of an international journal, it should be able to meet the interests of international readers. Therefore, there are still other aspects that need to be strengthened in this article.
The reviewers believe that with such a large amount of raw materials, it should be able to provide at least three papers.  Suggest the discussion part is still needs to be improve.

Author Response

Thank you for the comments and helpful suggestions regarding our manuscript. We have modified the manuscript accordingly and provided the revised details below.

Reviewer 2: "Less is more" is the motto of all things, especially for a complex conceptual essay. How to express its meaning in a simple and clear way is crucial for an excellent paper.
Thank the authors for theirs hard work to improve the quality of the content of this article.
This article has its own value for the development of Korean health industry. For the publication of an international journal, it should be able to meet the interests of international readers. Therefore, there are still other aspects that need to be strengthened in this article.
The reviewers believe that with such a large amount of raw materials, it should be able to provide at least three papers.  Suggest the discussion part is still needs to be improve.

Response: We apologize for the fact that we have not been able to significantly revise the discussion because there has been no prior research on patient safety literacy. I have added the below content in the manuscript.

  • Line 311- 315: In Korea, Patient Safety Offer conducted campaigns such as fall-prevention education and “Speak Up” for patient safety education. However, it has been reported that such patient safety education has limitations in preventing medical errors [14]. Therefore, it is necessary to develop standardized patient safety education materials that can help prevent medical errors in Korea. For this purpose, we developed patient safety education materials (“4 Tips for patient safety”) and patient safety literacy.